# CloudFCN: Accurate and Robust Cloud Detection for Satellite Imagery with Deep Learning

**Alistair Francis [1,2,][*][], Panagiotis Sidiropoulos [1,3][] and Jan-Peter Muller [1][]**

1   Mullard Space Science Laboratory, UCL, Holmbury Hill Rd, Dorking RH5 6NT, UK;
    panos@hummingbirdtech.com (P.S.); j.muller@ucl.ac.uk (J.-P.M.)
2   Cortexica Vision Systems Ltd., 30 Stamford Street, London SE1 9LQ, UK
3   Hummingbird Technologies Ltd., 51 Hoxton Square, Hackney, London N1 6PB, UK
*   Correspondence: a.francis.16@ucl.ac.uk; Tel.: +44-1483-204926

**Abstract:** Cloud masking is of central importance to the Earth Observation community. This paper deals with the problem of detecting clouds in visible and multispectral imagery from high-resolution satellite cameras. Recently, Machine Learning has offered promising solutions to the problem of cloud masking, allowing for more flexibility than traditional thresholding techniques, which are restricted to instruments with the requisite spectral bands. However, few studies use multi-scale features (as in, a combination of pixel-level and spatial) whilst also offering compelling experimental evidence for real-world performance. Therefore, we introduce CloudFCN, based on a Fully Convolutional Network architecture, known as U-net, which has become a standard Deep Learning approach to image segmentation. It fuses the shallowest and deepest layers of the network, thus routing low-level visible content to its deepest layers. We offer an extensive range of experiments on this, including data from two high-resolution sensors—Carbonite-2 and Landsat 8—and several complementary tests. Owing to a variety of performance-enhancing design choices and training techniques, it exhibits state-of-the-art performance where comparable to other methods, high speed, and robustness to many different terrains and sensor types.

**Keywords:** clouds; deep learning; machine learning; computer vision; multispectral; optical

## 1. Introduction

At any given time, around two thirds of the planet is obscured by clouds [1]. Hence, optical satellite instruments must contend with substantial cloud cover, obscuring the Earth's surface. For applications focused on surface processes—e.g., vegetation monitoring [2], or surveillance [3]—clouds must be accounted for and removed. Meanwhile, for applications that are directly related to atmospheric processes, such as weather monitoring, cloud pixels must be retrieved. Crucially, both removal and retrieval of clouds require pixel-scale segmentation of a satellite image into cloudy vs. clear regions. The large size of high-resolution satellite images, as well as the tedious nature of pixel-scale manual annotations, have been strong motivators for the development of fully automatic algorithms which aim to accurately and reliably detect clouds in satellite imagery.

Automatic cloud detection is not a straightforward problem. The expected accuracy is high, whilst oftentimes the ground-truth is ambiguous (especially at the boundaries of clouds [4]). As well as high accuracy, a successful algorithm must have low enough computational cost to be applied across large datasets, robustness over different terrains, and compatibility across different instruments and resolutions. Single-pixel techniques provide straightforward solutions which, if they use Machine Learning (ML), are easy to implement as fully differentiable models (differentiable models are ones for which all optimisation and parameter-tuning is governed by one loss function, as opposed to having

modular subsections of an algorithm which are optimised independently). However, these single-pixel methods do not utilise relationships between nearby pixels. Techniques which utilise high-level features can use spatial correlations to arrive at nuanced predictions when pixel-level information is not enough, but—as we will see in Section 2.3—they often use separate preprocessing stages, which are not optimised at training and can increase computational complexity. These approaches appear to be complementary, in that they achieve overlapping but non-identical parts of the set of required properties. Therefore, our hypothesis is that an approach which carefully combines pixel-level with spatial features in a single differentiable architecture could have the beneficial properties of rich multi-scale features, without the need for optimising different stages in isolation.

This work builds upon the aforementioned hypothesis, introducing a cloud detection pipeline that incorporates both low-level colour features and high-level spatial descriptors. To do this, we use a bespoke transforming autoencoder architecture, inspired by the U-net algorithm [5], which has recently demonstrated good performance in a variety of pixel-wise image segmentation tasks (e.g., [6,7]). The residual connections within the model allow for fusion between the low-level information of individual pixels and high-level convolutional features from successively larger fields of view. Meanwhile, the weights within the model's layers determine the specific features extracted, and their relative importance when fused, and are learned during a supervised training stage. Non-differentiable stages (e.g., hard-coded preprocessing pipelines) are avoided by giving the model the pixel values themselves as input, and outputting the final pixel-wise cloud masks at the end.

The design of our model is guided by theoretical considerations and intuitive reasoning based on the specific problem of cloud masking. More specifically, the concept of a 'receptive field' is used as a rationale for network size, which is very often a parameter simply tuned based on computational limitations rather than domain-specific knowledge. Similarly, our use of InceptionNet-style modules [8] within the network is informed by the observation that cloud edges are often ambiguous, and thus for a model to produce smooth outputted edges, local information sharing should be encouraged between neighbouring pixels. Finally, our class-weighting scheme provides both stable performance on this problem, but also to other segmentation problems in Remote Sensing, in which class populations are very often imbalanced.

Aside from the development of a high-performance algorithm, the main contributions of this study are the principles employed in designing, training and testing the algorithm. Specifically, our reasoning about the network's receptive field as a key design driver, and the class weighting approach we take (which are both applicable to image segmentation tasks in general). This study is a more complete and thorough version of preliminary work presented at an ESA conference [9], adding an extensive set of experiments on datasets totalling over 700 scenes and around 20 billion pixels. These are taken from multiple instruments, resolutions, geographical locations and terrain types, and are tested with several evaluation metrics, each focusing on a specific use case or performance criteria. Pixel-wise evaluation metrics (Accuracy, Omission and Commission rates) are used to compare performance with other cloud detection algorithms on well known datasets. Meanwhile, scene-level coverage accuracy (in which the total estimated cloud cover over an image is measured) is used to evaluate it's potential for on-board use. In addition, simulated noise experiments infer the algorithm's performance on instruments with a range of SNRs and bit-depths, whilst also shedding light on how the model uses textural and pixel-level features to mask clouds.

The rest of the paper is structured as follows: Section 2 summarises previous work on the topic, before the proposed CloudFCN algorithm is presented and analysed in Section 3. Subsequently, the more theoretical and detailed aspects of our work are discussed in depth in Section 4, while in Section 5 several experiments are documented. Section 6 summarises the implications of this paper and the future work it enables.

## 2. Related Work

### 2.1. Cloud Detection

Cloud detection in remote sensing is a long-standing problem for which many different solutions have been proposed. These are typically specific to instrument-type or spectral configuration. In this work, we focus on cloud detection algorithms applicable to single-frame visible and multispectral imagery only, hence, we do not examine (and do not compare in the evaluation section) multi-image cloud detection methods that rely on prior georeferencing information [10].

It should be noted that the detection of haze and ice fogs is not examined in this paper. Approaches that deal with haze or ice fog measurement or removal from images, as distinct from clouds [11–13], are of significant practical importance. However, they lie beyond the scope of this work, due principally to the lack of large labelled datasets containing haze or ice fog as distinct classes. Annotation of these features is not easy, given their similarities with cloud.

Spectral bands offer different and complementary information with which to detect cloud. For example, visible bands like blue, green and red exhibit high albedo on cloudy regions, which can be used to distinguish them from surface features which often have lower albedo in some or all of these bands [14]. However, some surface features exhibit high albedo in all visible bands (e.g., roofs or exposed rocks), and so bands such as SWIR can be used [15] as they are very sensitive to water and water vapour [16]. SWIR bands can also be helpful in distinguishing icy terrain from cloud [17], which is an issue that has been studied extensively [18]. Similarly, thermal bands—whilst lacking the high resolution of other bands—can measure an object's temperature, which for clouds is generally much lower than unobstructed surface regions.

Relevant cloud detection techniques can be divided into two main classes: thresholding approaches derived from physical or empirical observations, and Machine Learning (ML)-based approaches which rely on statistical reasoning. Thresholding algorithms have a set of rules based on the relative values of various combinations of bands, in order to evaluate whether a given pixel is clear or cloudy. These methods are often informed by physical parameters such as reflectance and/or top-of-atmosphere temperature (e.g., [19–21]). On the other hand, ML techniques take these spectral bands (or some function of them) as inputs, and through some optimization process derive a model that maps the input space onto the desired classes, with little to no dependence on the underlying physics.

### 2.2. Thresholding Techniques

In the early 1980s the need for automatic cloud masking was recognised, due to ever increasing quantities of EO data [22]. It was found in [23] that a thresholding technique based on radiative transfer analysis was significantly better when compared with other thresholding techniques based on clear-sky radiances. The radiative transfer method was successful because it computed two successive thresholding operations, in both visible and IR bands, making it more robust than techniques which relied on only one threshold. Since then, thresholding techniques have often taken the form of decision trees in which if-then nodes are computed using relational operators, on band values or derived physical parameters like temperature (e.g., [19,24–26]). Some algorithms then use post-processing techniques to refine the results. For example, Refs. [20,27] use the known illumination conditions of the scene to refine their estimates through geometric measures of similarity between detected clouds and shadows. Other techniques focus on classification of particular subcategories of cloud, such as cirrus clouds [28,29], or cumulonimbus clouds [30]. Spatial correlations between neighbouring pixels can also be used to improve classification accuracy and smooth the detected cloud boundaries, as exemplified in [31]. Recently, Ref. [32] classified cloud and clear pixels adjacent to cloud, motivated by the importance of performance on the cloud boundaries.

As noted in [33], the physical basis behind thresholding methods means that they can be straightforwardly transferred between different instruments and datasets, assuming a similar resolution and spectral range. Another advantage of thresholding techniques is that no training

dataset is required, which is time-consuming to produce. However, their simplicity is a significant limitation on their performance, as they are incapable of using more complex relationships between inputs, and are limited to instruments which provide the necessary spectral bands.

## 2.3. Machine Learning Techniques

ML algorithms differ fundamentally from "physical" cloud masking algorithms, in that they generate the function that maps input data to prediction by using statistical analysis of a training set. The number and arrangement of trainable parameters within the employed model dictates the space of possible solutions, whilst the training set used dictates the values of those weights and thus the specific solution found within that space [34]. Therefore, the variations seen in performance between cloud masking methods are effected by both the model architecture and training data.

Prior to the recent emergence of Deep Learning, most ML-based approaches focused on single pixels. Methods using individual pixel values as inputs to classification algorithms are well studied: with classifiers such as Support Vector Machines (SVM) [35], PCA [36] and classical Bayesian methods [29] being frequently tested with varying levels of success. Single-pixel neural networks (NNs) are also common among cloud masking algorithms (e.g., [31,37,38]). These methods benefit from their computational speed, because more complex relationships between neighbouring pixel values are not calculated. However, the spatial correlations that these single-pixel techniques ignore are important in most natural images, including remote sensing data. In essence, these single-pixel level techniques explore an input space of limited dimensionality (e.g., the number of spectral bands of the instrument), and cannot express functions radically different from those of physically-derived thresholding techniques.

Expanding the dimensionality of the input space by including multiple pixels introduces the possibility for more complex relationships than those which can be mapped from a single pixel, and can be expected to increase potential performance. However, the implemented model must be capable of extracting meaningful information from this larger input space. Several methods for extracting features from larger regions exist; single-band textural features have been used as inputs to classifiers [37], or in conjunction with pixel-level features [39]. Convolutional Neural Networks (CNNs) have had widespread success in image processing applications due to their ability to efficiently learn features across an extended scene (e.g., [40]). Commonly, these are used for classification tasks.

If pixel-wise segmentation is not desired, and area-wise classification is acceptable, then standard convolutional networks can be employed to classify square regions of an image, as in [41]. Another method for approximating cloud segmentation as a classification task is through superpixel construction. Superpixels can be used to break scenes up into a set of regions that contain similar pixel values [42]. This allows regions to be treated like discrete objects, to be classified by a CNN [43,44] or bespoke architectures like PCANet [45] as cloud or non-cloud. However, this can cause issues, as clouds are very often amorphous and merge with one another, meaning the performance is highly sensitive to the initial superpixel construction itself, which cannot be updated during training.

Ultimately, feature transforms like superpixel construction and textural transforms such as Gabor filters, are all non-differentiable pre-processing stages, and hence cannot be optimised for the specific task of cloud detection. This means they are unlikely to create the most suitable inputs for the vast majority of machine learning tools (e.g., a CNN, NN, SVM or Random Forests).

Recently, Fully Convolutional Networks (FCNs) [46] have shown promise in image segmentation problems across a variety of fields (e.g., U-net in biomedical imaging [5] and SegNet for natural image segmentation [47]), by outputting pixel-wise predictions over extended scenes. They do this by applying both convolutions and their inverse—transpose convolutions—to learn a direct transformation between image and outputted mask. This offers a framework for a model that successfully fuses pixel-level and spatial features. Perhaps the work nearest to our own is the one presented in [48], which used an FCN for cloud masking. Their algorithm includes a pre-processing thresholding step for snow/ice retrieval, and uses an existing Landsat 8 cloud mask as a prior to

improve performance. In a similar vein, ref. [49] use rescaling to combine features from different depths in a U-net.

Although promising, Deep Learning approaches have not yet demonstrated interoperability on multiple satellite sensor types without retraining [50]. This is a significant advantage of thresholding techniques, because they use physical knowledge to inform how different bands are used, whereas current ML architectures fail to integrate this physical knowledge into their decision-making (although this is an area of active theoretical work in ML e.g., [51]). So, despite the promising performance of ML models, this makes generalisation to instruments with other spectral bands very challenging.

Regardless of the architecture, supervised ML approaches require sufficient training data in order to achieve maximum performance. Although Transfer Learning can allow a model trained on one dataset to be used more easily on a new instrument, some training data is still inevitably required for each new application of the model. The community has in recent years made several annotated datasets publicly available, such as the 'SPARCS' dataset [52], comprising 80 1000-by-1000 pixel Landsat 8 patches, or the 'Biome' dataset [33], comprising 96 roughly 7000-by-7000 pixel Landsat 8 patches. For single-pixel approaches, Ref. [29] released a dataset of labelled spectra for Sentinel-2. With these datasets, among others, the potential for ML techniques in cloud detection can be explored more fully than was previously possible.

## 3. Materials and Methods

### 3.1. Overview

As mentioned in Section 2.3, pixel-level data necessarily has fewer dimensions than multi-pixel windows, which can make reliable detection of cloud more difficult. This is exacerbated in RGB when compared to multispectral data, because high albedo terrains (bright urban areas, for example) look the same in RGB as cloud pixels do, when they are be more separable in NIR and TIR bands. Although individual pixels can be misleading over certain terrains, texture often varies greatly between cloud and non-cloud. Therefore, if the system is to perform robustly in high albedo terrains, we need to extract spatial features around each pixel, as well as the individual pixel values themselves, and combine their information content to help distinguish cloud and non-cloud by their texture and surrounding context. Fusion of these different features is vital, because using only high-level features may make it difficult to learn simpler relationships based on colour and brightness. Using an FCN with residual connections in the style of U-net achieves these aims, whilst allowing the entire model to be optimised end-to-end during the training phase.

As input, CloudFCN takes extended scenes from satellite imagery with arbitrary spatial dimensions. For each given spectral band combination (e.g., RGB) a different instance of the model is needed, however the only alteration made to the design is the number of input channels—no internal parameters are altered. The output of the model has the same spatial size as the input, but can take two distinct formats. When a pixel-by-pixel mask is desired, the final layer uses a softmax activation [53] to classify each pixel as Clear or Cloudy. This is advantageous because it can be easily extended to multi-class problems (e.g., Clear vs. Cloud vs. Cloud Shadow). However, when cloud coverage estimation (as in the percentage of cloud cover over a whole scene) is the desired final product, we output a mask where each pixel has a value between 0 (clear) and 1 (thick cloud), and the cloudiness of a pixel is treated as a regression problem. We then take an average over the whole scene for the cloud coverage estimation. This format allows the model to make estimates for thin cloud as well as thick, and results in more accurate percentages through averaging.

The model comprises two arms, the *encoder* and the *decoder* (Figure 1). The convolutional structure of these arms is described further in Section 3.2. The encoder serves to extract spatial features from the scene, reducing the spatial dimensions whilst increasing the number of channels. Meanwhile, the decoder takes these features and reprojects them to create the output mask. Residual connections

(also described in Section 3.2) link intermediate points in the encoder and decoder, to allow for fusion between low- and high-level features.

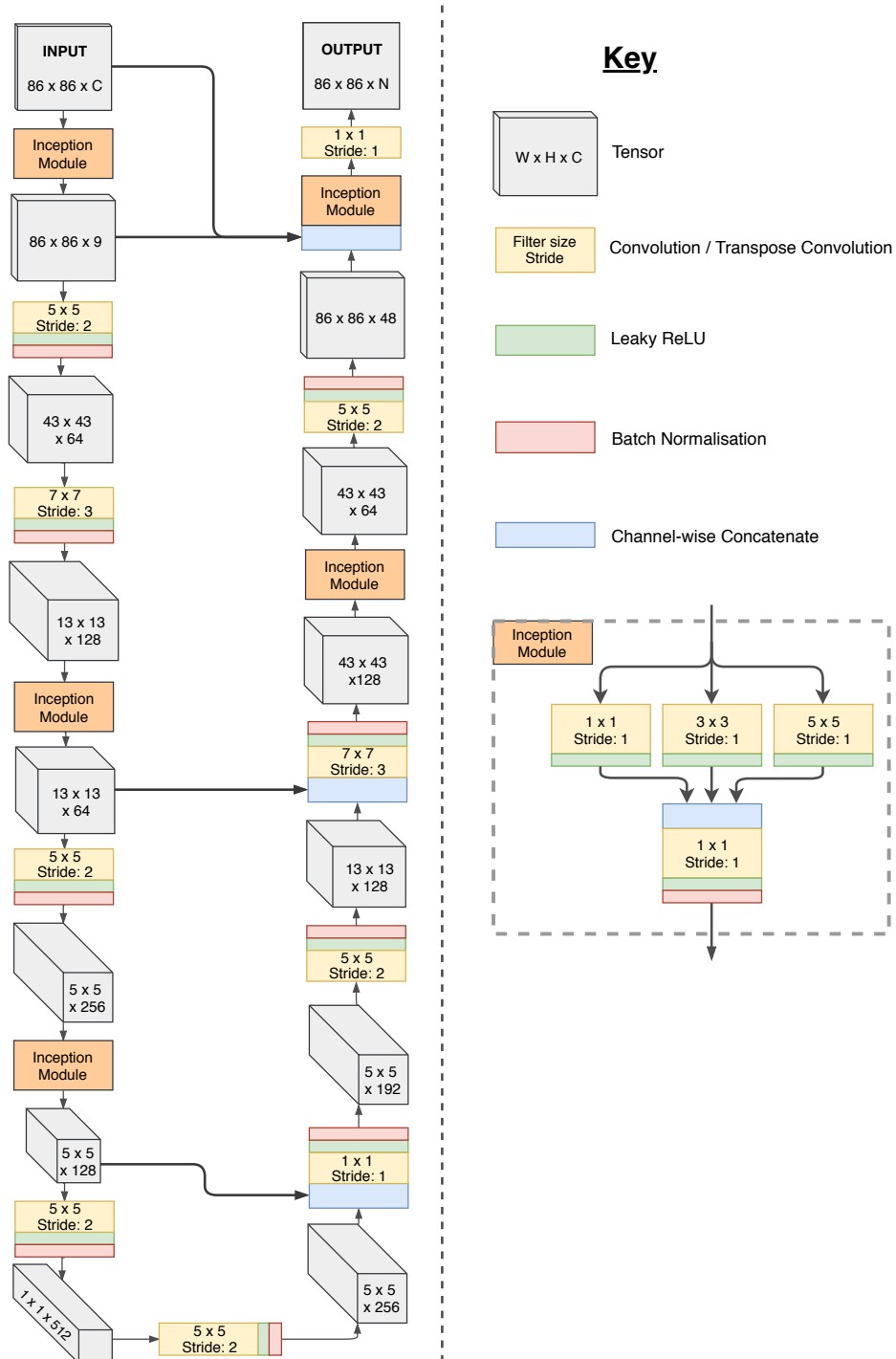

**Figure 1.** Flowchart of CNN used in cloud segmentation. Bold arrows represent residual connections, connecting different stages of the encoder and decoder arms of the model which run in a U shape. 'Inception module' denotes a set of parallel convolutions of stride 1, followed by a single dimensionality-reduction convolution on their concatenated outputs. Widths and heights of inputs and outputs are the *minimum* and could be any integer multiple of 24 above 86, with internal tensor dimensions changing accordingly. The first input layer is given C channels, as a placeholder for whatever kind of input being used (e.g., 3 for RGB). The output has channel depth N, to denote the number of output classes used.

The proposed model has several desirable traits, which alleviate the issues that other methods exhibit. First, our design allows for flexible input formats, having no specific spectral requirements, and is able to ingest input images of different sizes. Second, we combine the simplest brightness and colour information with more abstract and complex features from the surrounding areas, creating a complementary set of features that exhibit the strengths of both pixel-level techniques and convolutional ones. Lastly, the output format of the model, and attendant loss function, can be selected based on user needs, increasing the number of possible applications for the method, from cloud coverage estimation (in which a RMSE loss is used) to pixel-wise masking (with a categorical-crossentropy loss).

### 3.2. Model Features

The convolutional layers of the model are designed to extract translation-invariant features across the image. Each filter within a layer is convolved with the input, producing a response map of where the filter pattern is similar to locations in the image. The advantage of convolutional layers is their shared weights—by using the same filters across the whole image, we reduce the number of weights whilst enforcing translational invariance of the learnt features. Additionally, using exclusively convolutional layers is what allows us to create models that work on arbitrary image sizes, because no layer requires a specific input size, unlike fully connected layers commonly seen in classification networks. By using convolutions of stride greater than one, the layers' outputs are spatially downsampled, leading to more compact feature map representations.

Between the standard convolutional layers, which reduce the spatial dimensions of the data whilst increasing the channel depth, we include InceptionNet-style modules [8] with several parallel convolutional layers of stride one, all with different filter sizes. These layers preserve the spatial dimensions of the feature map and fuse information from the surrounding area at each point (Figure 1). As well as allowing for connections between neighbouring points, these modules also reduce the channel depth of the feature map, which increases the computational efficiency and reduces memory requirements. We also found that using these layers (particularly the one connected closest to the output) reduced the amount of high frequency checkerboard artifacts in our output. These are seen commonly in transforming encoders, and are a result of uneven responses from the final transpose convolution layer [54].

Residual connections are employed in our model to allow for different feature levels to be fused at various points in the decoder arm. By fusing the features, we ensure that the final output is informed by both simple intensity relationships between different spectral bands in the individual pixel, as well as by a range of complex spatial features. The residual connection concatenates the features outputted by one of the encoder's convolutions onto the inputs of one of the decoder's transpose convolutions.

Before three of the decoder's transpose convolutions, there exists a residual connection. This effectively means there are 4 nested transforming encoder routes (3 across the residual connections and the last through the entire encoder and decoder) which are all used to arrive at an output value for a given pixel. Each of the 4 transforming encoder routes provides information about a different field of view, the first being pixel-level, the second a small window around each pixel, the third a larger window, and the last having the maximum field of view of the network. The field of view of a given layer is termed the *receptive field*, and is discussed in more detail in Section 4.1.

After each convolution in the network, we use a leaky ReLU activation function, which introduces non-linearities [55]. A leaky ReLU is similar to a standard ReLU layer, in that it has a gradient of 1 for inputs above zero, but, unlike the flat response of a ReLU for input values below zero, it has a small gradient (Equation (1)). This allows negative values to still have a gradient in back-propagation, preventing them from getting 'stuck' at some value.

$$y = \begin{cases} x, & \text{if } x \geq 0 \\ ax, & \text{otherwise} \end{cases} \quad, \quad where\ 0 < a \ll 1 \tag{1}$$

Batch normalization layers [56] are used after each convolution or inception module. These rescale the response of a layer to have a mean of zero and a standard deviation of 1. Batch normalisation has several advantages. Primarily, it acts to regularise the network, thus reducing overfitting, which is especially important when using relatively small datasets like those available in cloud masking. It also helps to increase training speed, as gradients are re-scaled at each layer during back-propagation.

We opted not to use max pooling layers [53]. During max pooling, the feature map is downsampled by selecting the maximum value in each channel within a given window size, (typically 2-by-2, leading to a downsampling rate of 2). By doing so, one condenses the feature map and preserves the most important features, but information about where exactly those feature are present is lost. For classification tasks, this is not so important, however when the output is a pixel-wise mask, this loss of localisation could lead to inaccuracies in the output.

The number of layers within the model and their geometries were arrived at heuristically, by considering both the computational efficiency at inference and training time, as well as the model's performance. With a larger number of channels in each layer, the complexity of information that the model is able to describe increases. However, at arbitrarily high channel depth, the information content is limited by the available dataset and training time, rather than by the model's architecture. Therefore, we first arrived at a sensible number of layers (described further in Section 4.1). Then, the channel depth of each layer was selected by starting with a small number of weights, and gradually increasing them until performance was at a plateau.

### 3.3. Training

During training, we split the dataset into training and validation sets. We avoid training separate models per terrain, as we wish to aim for robust models that are invariant to terrain type. Therefore, we ensure both training and validation contains a representative sample from each terrain type. Although arbitrarily sized images can be ingested by the model, we train with image patches of a fixed size to allow for simpler implementation. Tests on the effect of cropped image size were inconclusive, suggest that anything larger than around 150 pixels had similar performance, whilst crop sizes at the minimum (86 pixels) made performance suffer slightly. At larger sizes (roughly over 500 pixels), the memory requirements for back-propagation over large batches became unfeasible on our 6GB GPU. Therefore, we mostly trained our models on patches of a few hundred pixels across. Importantly, although a training sample is a cropped patch from the original image, training and validation sets do not share parent images, in that they are split prior to any cropping operations.

Each batch contained 24 patches taken from the training set. The intensities are normalised such that across the entire dataset there is a mean = 0 and standard deviation = 1 in each spectral band. Next, each image patch goes through several random transformations—listed below—in order to increase the variance seen in the training set, and to ensure the model is invariant to moderate noise and small changes in illumination condition.

1. **Rotation:** Rotated by a random integer multiple of $90°$
2. **Flipping:** Flipped left-right with 50% probability
3. **Intensity shift:** All input values scaled by same random factor in range 0.9–1.1
4. **Chromatic shift:** Each input channel scaled by different random factor in range 0.95–1.05
5. **Salt and pepper noise:** Pixels set to hot (+3) or cold (−3) values, with per pixel probability of 0.005
6. **White noise:** Gaussian noise with sigma = 0.05

The loss function is determined by the output format. Two cases exist, in the case where cloudiness is treated as a continuous variable where 0 is clear and 1 is thick cloud, then mean square error is used in order to train for pixel-wise regression. For pixel-wise classification, we use categorical cross-entropy as the loss function. This loss is used by the optimisation routine to update weights. We experimented with Stochastic Gradient Descent (SGD) [57], Adam [58], AdaGrad [59] and AdaDelta [60]. All three

Adaptive update methods outperformed SGD by similar amounts with regards to training time and final performance, so Adadelta was selected somewhat arbitrarily.

Training was carried out until no notable improvement in performance on the validation set was seen over several epochs. Generally, a few thousand batches were used, taking under an hour on a desktop GPU (NVidia GTX 1060, 6GB) to reach a plateau. A sample of validation results were also visually displayed in each epoch, to gain a qualitative appreciation for the model's performance, and the characteristics of the output masks.

## 4. Theoretical Considerations

### 4.1. Receptive Field

The receptive field is defined as the region of inputs that can affect the value of a given node in the model, outside of which any variation in input would have no effect on that node's response. In designing the model architecture, we first intuited a reasonable value for the receptive field of the output pixels. Ultimately, the optimal value is not exact, but lies in a range such that it is neither too small or too big. If it is too small, then the network will not have access to useful information about a pixel's surroundings, and if it is too big then the network's size becomes cumbersome and training time, computational power and dataset size limits performance. So, to find a sensible range we qualitatively assessed the ability of humans to recognise cloud in cropped Earth Observation images. At very small crop sizes (a few pixels) we as humans are often unable to discern between cloud pixels and other high-albedo regions, because contextual information about the scene is not given to us. At very large crop sizes, humans do not improve in their cloud masking predictions, because the limit of useful contextual information is reached. We reasoned that a field of view of a few hundred pixels was ample for humans to make predictions of cloud masks, based on our experience observing several Landsat 8 scenes. Therefore, we designed the model such that the receptive field of its final convolutional layer was roughly 300–400 pixels across. Going beyond this value would lower the computational efficiency, making it difficult to process larger scenes quickly.

Receptive fields can be calculated recursively throughout the layers of the network, beginning at the input [61]. First, we consider the receptive field, $r_{i+1}$, of a neuron in *layer*$_{i+1}$ given the geometry of *layer*$_i$ (stride, $s_i$ and kernel size, $k_i$) (Equation (2)). In addition to the previous layer's parameters, the jump, $j_i$ is defined as the product of all strides up to that layer, and is used to keep track of the 'distance' in input space between two adjacent points in *layer*$_i$ (Equation (3)).

$$r_{i+1} = r_i + (k_i - 1) * j_i \tag{2}$$

$$j_{i+1} = j_i * s_i \tag{3}$$

Table 1 shows the receptive field and jump through successive layers in the network. Up until the *code layer*, which signifies the deepest part of the network (Figure 1), the jump increases along with the receptive field. A neuron in the code layer of the network describes a feature that is sensitive to a 165-by-165 pixel area. The jump of 24 means that for every 24 pixels in the input image, one more feature vector is used in the code layer. After this point in the model the jump begins to decrease, whilst the receptive field continues to grow. This is because the stride of a transpose convolution has the inverse effect of a convolutional layer on the jump.

Although the total receptive field of an output pixel is 365, the outermost regions will only negligibly impact the model's response. This is partly an inherent property of many-layered convolutions [62], but is in our case amplified by the use of residual connections. Each residual connection adds more importance to a central portion of the receptive field. For example, the first residual connection adds only pixel-level information, thus amplifying the importance of only the central pixel in the receptive field, whilst a residual connection deeper in the model will amplify the importance of a 21 or 69 pixel region at the centre of the full receptive field. This is a desired property,

as it is clear that the importance of a feature 100 pixels away is less than that of a pixel in the output neuron's proximity.

**Table 1.** Layer-by-layer calculation of receptive field through the network. This follows the encoder and decoder displayed in Figure 1. The jump reaches its maximum before the first transpose convolution, where it then begins to decrease. Blue rows indicate that their outputs are sent through a residual connection, whilst red rows indicate the operation has a residual connection to its input.

| Layer | Layer Type | Stride | Filter Size | Jump | Receptive Field |
|-------|-----------|--------|-------------|------|-----------------|
| 1 | Inception | 1 | 5 | 1 | 1 |
| 2 | Convolution | 2 | 5 | 1 | 5 |
| 3 | Convolution | 3 | 7 | 2 | 9 |
| 4 | Inception | 1 | 5 | 6 | 21 |
| 5 | Convolution | 2 | 5 | 6 | 45 |
| 6 | Inception | 1 | 5 | 12 | 69 |
| 7 | Convolution | 2 | 5 | 12 | 117 |
| 8 | Transpose | 2 | 5 | 24 | 165 |
| 9 | Convolution | 1 | 1 | 12 | 261 |
| 10 | Transpose | 2 | 5 | 12 | 261 |
| 11 | Transpose | 3 | 7 | 6 | 309 |
| 12 | Inception | 1 | 5 | 2 | 345 |
| 13 | Transpose | 2 | 5 | 2 | 353 |
| 14 | Inception | 1 | 5 | 1 | 361 |
| 15 | Convolution | 1 | 1 | 1 | 365 |
| 16 | Output | - | - | 1 | 365 |

*4.2. Class Weighting*

Imbalanced class populations in a training set can lead to unwanted characteristics in the resulting model. In the extreme case, where one class represents a huge majority of training examples, the loss function drives the model to always predict the majority class, and never consider the rare minorities. Even when moderate class imbalances are present, the model can still develop strong biases against rarer classes. This *a priori* bias, although good for minimising loss on training data, may not be desirable in real-word use. For example, if overall accuracy is encouraged by the loss function, then the precision of a rare class will be prioritised over the recall, as the model will be more sceptical. For problems in which high recall is desired, this is an unwanted effect.

In many ML applications, imbalanced datasets can be addressed by preferentially sampling minority classes at training time. Alternatively, the loss associated with each class can be modified by a factor which prevents the model from becoming overly biased. In our work we opted to modify the loss by a factor that normalised for the relative abundances of the different classes. In the Landsat dataset used, cloudy and clear pixels took up relatively equal amounts of the data, so this was not as important. However, when used for Carbonite-2 data, the number of cloudy pixels was too low and a strong bias against predicting cloud was found in the model. Therefore, loss weighting was applied to encourage the model to predict the presence of cloud.

We found a factor of $1/abundance$ applied to the loss was too dramatic (where *abundance* is defined as the proportion of total pixels that are within that class), as for cloud this would mean a loss that was nearly an order of magnitude larger than for clear pixels, leading to unstable back-propagation. Instead, we used a factor of $1/\sqrt{abundance}$ which was a trade-off between balancing Omission and Commission errors and producing gradients small enough to lead to stable back-propagation. Formal experimentation on the amount of class-weighting was not conducted, because its purpose is not to fine-tune the Omission and Commission errors, but rather to ensure stable training that consistently finds a suitable local minimum. A more practical way to tweak the balance between Omission and Commission is to adjust the threshold confidence value at inference time, in order to optimise the performance for a given application.

## 5. Experimental Results

The performance of CloudFCN is measured in several experiments, with metrics that highlight different properties of the model. In Section 5.1, we describe the two datasets used in our experiments. Then, in Section 5.2 an experiment for cloud coverage estimation in RGB is conducted, which is relevant to possible on-board use of the algorithm. Next, in Section 5.3, pixel-wise cloud masking performance is measured on a Landsat 8 dataset, with only RGB bands taken from the 11-band instrument. Section 5.4 then tests the performance of the algorithm on full 11-band Landsat 8 images, and offers a comparison between several algorithms and our own. Finally, in Section 5.5, we analyse how noise affects the algorithm's performance, by corrupting Landsat 8 data and re-evaluating the experiments performed previously. Evaluation of the algorithm's performance in cloud shadow detection is omitted, because the available datasets did not have consistent shadow annotations (the Biome dataset does have some labelled cloud shadows, however it is not comprehensive, making training and validation on this class challenging).

### 5.1. Datasets

#### 5.1.1. Carbonite-2

The Carbonite-2 satellite takes high-resolution (80 cm per pixel) video in visible wavelengths. By gimballing the platform as it orbits, a continuous video over a single scene is taken, at a size of 5000-by-5000 pixels. The dataset we have created comprises individual frames from several hundred videos from a variety of locations around the world (Figure 2). From each video used, we selected 3 frames randomly and marked both 'thick' and 'thin' cloud. 'Thick' cloud we define as any cloud cover through which no surface features are visible, whilst 'thin' cloud is any cloud through which some surface content is visible.

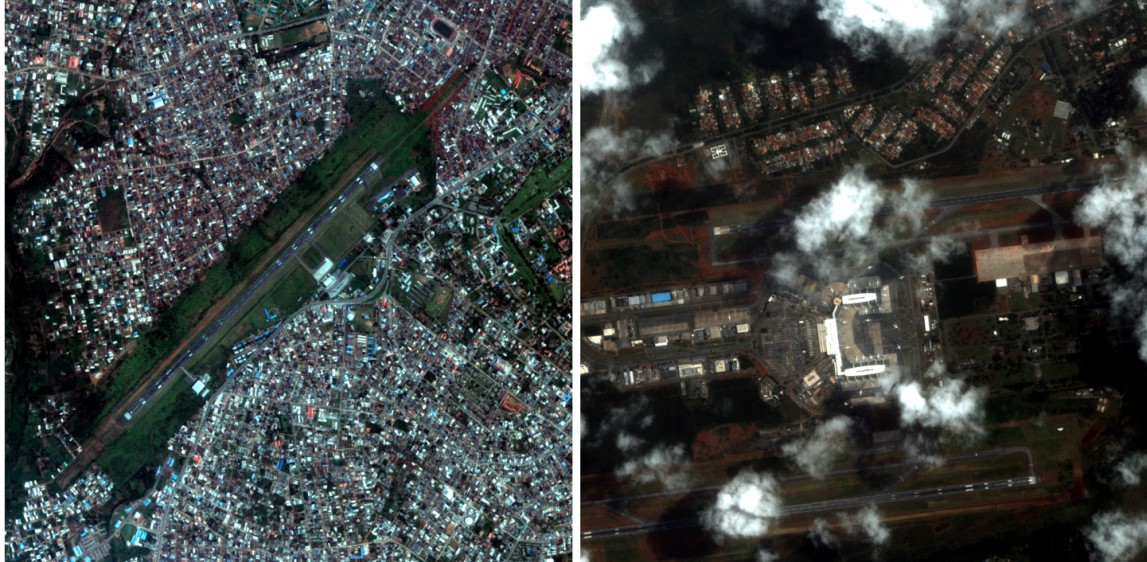

**Figure 2.** Two examples of data from the Carbonite-2 satellite. The left-hand pane shows a scene from Benin, and the right-hand pane is over an area in Brazil. Data provided by Surrey Satellite Technology Ltd. and Earth-i Ltd.

In total, 1561 frames were annotated, however, many of these (698) contained large amounts of noise, and were unusable. This left 863 frames to be used in training and validation. The vast majority of frames selected were entirely clear, or entirely cloudy, as these were the simplest to annotate. However, 155 frames were manually annotated and contained both clear and cloudy conditions. The training set was made using all the mixed visibility frames along with some all cloud and all clear frames, totalling

185. For validation, we conducted tests on both all cloudy and all clear frames. 72 all cloud frames were left after making the training set, whilst 606 all clear ones were. This validation strategy focussed on assessing the model's reliability as a cloud coverage estimator, rather than as a masking algorithm. This means we were more interested in severe errors in which an all clear frame was marked as mostly cloudy, or vice versa. We leave the analysis of pixel-wise predictions for the other dataset.

5.1.2. Biome

The Biome dataset [33] (Available from: https://landsat.usgs.gov/landsat-8-cloud-cover-assessment-validation-data) is a collection of level-1 Landsat 8 images. Comprising 96 individual scenes, this dataset was specifically designed for testing cloud masking performance in a variety of terrain types. In total, 8 categories of terrain are given, with 12 tiles each: Barren, Forest, Grass/Crops, Urban, Shrubland, Snow/Ice, Water, and Wetlands. The annotations have classes for clear, shadow, thin cloud and thick cloud. However, the creators note that shadow was inconsistently marked due to difficult topography, and we found it too sparsely annotated to be useful in training. Therefore, we condense the original 4 class problem into a two class problem: clear vs. cloud, in which clear includes shadows and cloud is the combination of thin and thick annotations. This classification scheme allows for direct comparison with the 9 algorithms tested with the dataset in [33] for cloud detection.

When training and validating on the Biome dataset, we opted to split the dataset into two halves, each with 6 of the 12 scenes from each biome, and with similar average cloud percentages. For each experiment, the model is initialised randomly and trained on one half, then validated on the other. This is repeated with the halves switched, and the statistics are aggregated between the two.

*5.2. Cloud Coverage Estimation*

In this experiment, we aim to show how the algorithm can be used as a simple coverage assessment tool, rather than as a pixel-wise masking technique. This is important to test for several reasons. First, by checking accuracy over extended images, we are judging whether the algorithm has strong systematic errors over certain terrain or cloud types, which would lead to dramatic biases over certain scenes. Second, on-board data quality assessment would likely be more reliant on cloud coverage, in order to make operational decisions as to which images to downlink.

Table 2 shows the results of the coverage estimation experiment. The subclasses of images were chosen not only to further detail the performance of CloudFCN in different situations, but also give insight into CloudFCN's properties. Urban areas and airports are characterised by many small high albedo areas (rooftops and concrete areas) which CloudFCN could distinguish from cloud by their surrounding texture, rather than by their colour. Meanwhile, waves have both similar colour and texture to cloud, making them a challenging surface feature to discriminate, which was reflected in the results. Snow-Ice was also understandably challenging, however complete failures (>50% error) were still rare.

The Cloudy class is somewhat difficult to interpret. In fact, very few frames were found that were entirely cloud. Most of the frames used had small regions without cloud, making predictions of high cloud cover less likely by the model. Therefore, only the most egregious error populations (>20% and >50%) are relevant.

Overall, outlier rates are low, meaning there are only a few example scenes that cause complete failure. This is a desirable property for the algorithm if it is to be used for on-board for data reduction, as very few cloudy scenes will be kept, or clear scenes discarded. Based on this experiment, only 4.16% of cloudy frames would be transmitted, at the cost of 0.66% clear frames being discarded.

**Table 2.** Full results of the Carbonite-2 experiments. Mean and median columns give respective error percentages. '>*x*%' columns give percentage of frames for which the error was higher than *x*. The results show a significantly skewed error population, with mean error consistently much large than median, showing that for the half the frames the accuracy is exceptionally good (<0.28%).

|  | Total | %Mean | %Median | >2% | >5% | >10% | >20% | >50% |
|---|---|---|---|---|---|---|---|---|
| Clear | 606 | 1.85 | 0.28 | 10.40 | 4.46 | 3.47 | 2.48 | 0.66 |
| Clear + Airports | 129 | 1.17 | 0.34 | 8.53 | 4.65 | 1.55 | 0.78 | 0.00 |
| Clear + Roads | 322 | 0.87 | 0.29 | 8.70 | 4.04 | 0.93 | 0.31 | 0.00 |
| Clear + Waves | 42 | 3.71 | 0.69 | 30.95 | 7.14 | 7.14 | 7.14 | 0.00 |
| Clear + Urban | 541 | 1.01 | 0.29 | 8.50 | 2.77 | 1.66 | 0.74 | 0.00 |
| Clear + Crops | 379 | 0.98 | 0.28 | 7.39 | 1.85 | 1.32 | 1.06 | 0.00 |
| Clear + Water | 196 | 1.59 | 0.40 | 12.76 | 4.59 | 3.06 | 2.04 | 0.00 |
| Clear + Snow-Ice | 62 | 9.34 | 0.29 | 24.19 | 19.35 | 19.35 | 17.74 | 6.45 |
| Clear + Docks | 79 | 1.94 | 0.40 | 10.13 | 3.80 | 3.80 | 3.80 | 0.00 |
| Cloudy | 72 | - | - | - | - | - | 12.50 | 4.16 |

### 5.3. Pixel-Wise Cloud Detection

This experiment is conducted to ascertain the detection performance of CloudFCN with RGB bands as input. Using the Biome dataset, we can isolate the performance in different kinds of terrain. The metrics used in evaluating our algorithm's performance are chosen to be the same as those used in previous studies [32,33] and are derived with respect to pixel counts: *cloud_as_cloud* is the number of cloud pixels predicted as cloud, whilst *cloud_as_visible* is the number of cloud pixels predicted as visible, and so on. *total_cloud* and *total_visible* refers to the total count of cloud and visible pixels respectively, and *N* is the total number of pixels. We use the term 'visible' to describe the union of both clear pixels and shadow pixels, given that we do not train the model to distinguish between them in this study.

$$\%Correct = \frac{cloud\_as\_cloud + visible\_as\_visible}{N} \tag{4}$$

$$\%Omission = \frac{total\_cloud - cloud\_as\_cloud}{total\_cloud} \tag{5}$$

$$\%Commission = \frac{visible\_as\_cloud}{total\_visible} \tag{6}$$

The final performance metric is given as the $\%Quality = \%Correct - \%Omission - \%Commission$ and is used to judge the algorithm's performance against others. The first rows of Table 3 documents these results for each of the 8 terrains in the Biome dataset, and the average over the dataset. Accuracy averaged 82.81%, although performance varied dramatically in different terrains. In Snow-Ice, accuracy was at 50%, primarily due to a high Commission rate which suggests it failed to learn features that separated snow from cloud. However, for 4 of the 8 biomes accuracy was above 92%. Using only visible bands, we outperform several previous algorithms which use multiple infrared channels. This lends strong evidence to the assertion that using multi-scale features allows for better performance in cloud masking.

The performance of CloudFCN in RGB seems highly dependent on the surface texture. Two of the worst performing terrains: Barren and Snow-Ice, are both characterised by large regions with relatively little texture. In other terrains, such as Urban or Forest, high frequency spatial features make the surface more distinguishable from the cloudy regions. Record performance was measured on Shrubland terrain—although it is not clear why this is the case—this does suggest that still larger datasets are needed in order to reliably gauge the model's performance.

**Table 3.** Cloud detection results for Biome dataset. RGB gives performance of algorithm with Landsat 8 bands 4,3,2 as input. Multispectral uses all 11 Landsat 8 bands. Comparison with 9 other algorithms validated in [33] are given at the bottom, with Quality values derived from the values given in the study. For each biome, the best-performing algorithm is in bold. The multispectral performs best overall, with an average quality metric 6.7% greater than the next highest, AT-ACCA.

| | | Barren | Forest | Grass-Crops | Shrubland | Snow-Ice | Urban | Water | Wetlands | Mean |
|---|---|---|---|---|---|---|---|---|---|---|
| **CloudFCN (RGB)** | %Correct | 78.92 | 82.69 | 94.69 | 93.41 | 49.65 | 93.41 | 92.34 | 77.36 | 82.81 |
| | %Omission | 12.24 | 23.20 | 4.08 | 8.82 | 7.25 | 2.88 | 6.34 | 24.32 | 11.14 |
| | %Commission | 29.89 | 11.00 | 7.36 | 4.47 | 76.96 | 8.64 | 8.18 | 19.90 | 20.80 |
| | %Quality | 36.79 | 48.48 | 83.25 | **80.11** | −34.56 | 81.89 | 77.82 | 33.14 | 50.87 |
| **CloudFCN (Multispectral)** | %Correct | 92.95 | 95.12 | 96.12 | 88.68 | 72.93 | 95.56 | 95.43 | 91.24 | 91.00 |
| | %Omission | 4.70 | 7.21 | 6.27 | 19.66 | 17.60 | 2.21 | 4.62 | 13.16 | 9.43 |
| | %Commission | 8.95 | 1.92 | 1.79 | 3.23 | 27.53 | 5.75 | 4.50 | 3.89 | 7.19 |
| | %Quality | **79.30** | **85.99** | **88.06** | 65.79 | 27.80 | **87.61** | **86.31** | 74.19 | **74.38** |
| **ACCA** | %Quality | 63.02 | 68.69 | 62 | 60.47 | **36.25** | 68.33 | 71.43 | 62.48 | 61.56 |
| **AT-ACCA** | %Quality | 66.67 | 73.83 | 74.09 | 70.65 | 35.86 | 74.06 | 70.51 | **76.25** | 67.72 |
| **cfmask** | %Quality | 77.1 | 67.27 | 85.74 | 75.53 | 26.37 | 74.72 | 50.98 | 65.97 | 65.69 |
| **cfmask-conf** | %Quality | 66.78 | 66.72 | 83.59 | 72.3 | 20.75 | 76.54 | 51.11 | 67.45 | 63.63 |
| **cfmask-nt-cirrus** | %Quality | 54.23 | 57.2 | 70.71 | 71.58 | −15.87 | 74.37 | 50.23 | 47.16 | 51.62 |
| **cfmask-nt-cirrus-conf** | %Quality | 54.44 | 38.79 | 60.01 | 66.38 | −43.68 | 73.2 | 49.04 | 35.14 | 41.66 |
| **cfmask-t-cirrus** | %Quality | 69.82 | 64.78 | 77.98 | 72.75 | −24.1 | 72.42 | 57.21 | 53.27 | 49.01 |
| **cfmask-t-cirrus-conf** | %Quality | 69.37 | 43.99 | 77.76 | 72.34 | −52.76 | 74.72 | 57.24 | 52.14 | 49.63 |
| **See5** | %Quality | 54.19 | 51.88 | 42.15 | 42.46 | 35.48 | 57.4 | 39.35 | 68.17 | 49.17 |

## 5.4. Multispectral Performance

Many instruments, including Landsat 8, collect data over a range of visible and infrared bands. This section will show how the inclusion of this multispectral data effects the performance of our algorithm. We repeat the experiments from Section 5.3 using all 11 bands of Landsat 8. Alongside these, we provide a comparison between our modes and several algorithms previously validated on the Biome dataset in [33]. Training and validation are done in an identical way to the previous experiments, and the same performance metrics used.

We found that the performance of the algorithm improved almost universally when all bands were used (see Figure 3). In 7 of the 8 terrain types within the Biome dataset, performance improved, often significantly (see Table 3). The most dramatic improvements were seen in Snow-Ice, and Barren terrains. These terrains also saw the worst performance in RGB, perhaps owing to their high reflectivity and relatively smooth textures, making them difficult backdrops for cloud detection. On average, the multispectral algorithm performed significantly better than all other tested techniques. It is worth noting, however, that the RGB CloudFCN still outperformed several of the previous algorithms.

It is not straightforward to posit which multispectral bands helped the model most in its predictions, given the black box nature of NNs. However, future work could explore more spectral combinations than those tested here, to better constrain which bands are useful for an architecture like CloudFCN. Nonetheless, the importance of spectral band selection in instrument design is underlined by this experiment. Not only does average performance improve substantially, but the variance in performance over different terrains is reduced by including more spectral data (from a range of 45% in RGB accuracies to a range of 23% in multispectral accuracies across the different terrains).

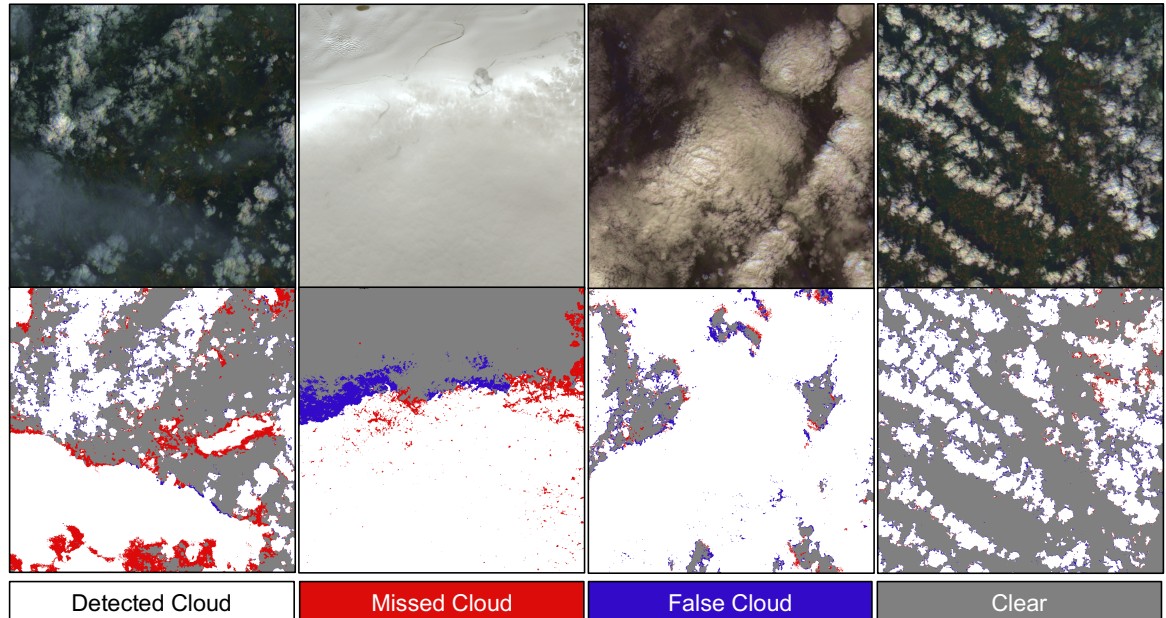

| Detected Cloud | Missed Cloud | False Cloud | Clear |

**Figure 3.** Four examples of cloud detection on the Biome dataset from the multispectral model. Very few mistakes are seen in the clouds' interiors, but the edges are more error-prone. The first example shows what may be a cirrus cloud over a vegetated region, the boundaries of which show noticeably more errors of omission than neighbouring cumulus clouds. The cloud over snow in the second pane is successfully detected, although large errors are seen at it's edge. The final two panes show highly successful detections of different kinds of cloud over varied terrain, despite some disagreement at the very edges of clouds, which are often not well defined and rather ambiguous. All scenes are shown as RGB composites using bands 4, 3 and 2.

*5.5. Noise Tolerance*

By testing on both Carbonite-2 and Landsat 8 data in several configurations, we have aimed to provide evidence that our algorithm is capable of high performance across a range of sensors. Remote Sensing cameras have a range of different sensitivities, bit depths and noise characteristics, which can adversely effect the quality of cloud masking algorithms. This section further proves the algorithm's generality by showing the effect on performance of adding white noise and quantization to our validation data. By testing whether our model is sensitive to these parameters, we can provide further evidence for it's general applicability as a cloud masking algorithm for Earth Observation.

Importantly, we are not interested here in directly assessing the performance on a noisy Landsat scene, as those scenes that are degraded badly by noise will not be used in applications anyway. Rather, these experiments provide us with a reasonable assessment of what level of noise we can expect our technique to overcome when used with other sensors. Comparison between the noise types also indirectly provide insight into the importance of pixel-level vs. textural features for CloudFCN's predictions, because quantization primarily effects texture whilst leaving pixel intensities relatively unchanged, whilst white noise affects pixel intensities more strongly.

5.5.1. White Noise

For the white noise experiment, we add Gaussian noise of varying powers to the Biome dataset. We do not retrain the model with more noise applied to it's training examples, instead just reusing the exact models trained for Table 3. For the purposes of this experiment, we measure only the white noise synthetically added by us, and do not include any term for the noise already within Landsat 8 images, as it is small in comparison to the added noise. The SNR (see Equation (7)) is used to measure

the relative power of noise to the original image's signal, and is defined with respect to the image's mean ($\mu_{signal}$) and the standard deviation of the noise ($\sigma_{noise}$).

$$SNR(dB) = 10log_{10}(\mu_{signal}^2 / \sigma_{noise}^2) \qquad (7)$$

We test our algorithm at SNR values ranging from 20 dB down to 7 dB, examples of images at different points in this range can be seen in Figure 4a. The Omission and Commission rates for both RGB and multispectral models on the Biome dataset can be seen in Figure 5a. In general, Omission increased strongly with noise level, whilst Commission was less affected, even going down somewhat for the RGB model. RGB Omission was more sensitive, increasing from 11.0% by 42.2%, when compared to multispectral Omission increasing from 9.4% by 18.2%.

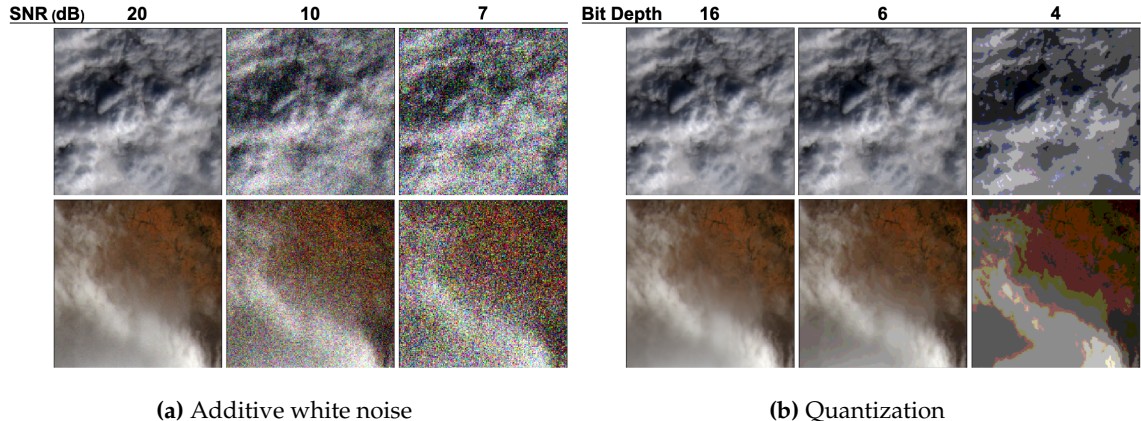

**(a)** Additive white noise      **(b)** Quantization

**Figure 4.** Examples of the effect of white noise and quantization throughout the range of levels applied to the Biome dataset for the validation in Section 5.5. For white noise, an SNR of 7 dB represents a significant distortion of the data, leading to most small features being over-powered by the white noise. Quantization also leads to a loss of textural information, acting to smooth large areas with similar intensities.

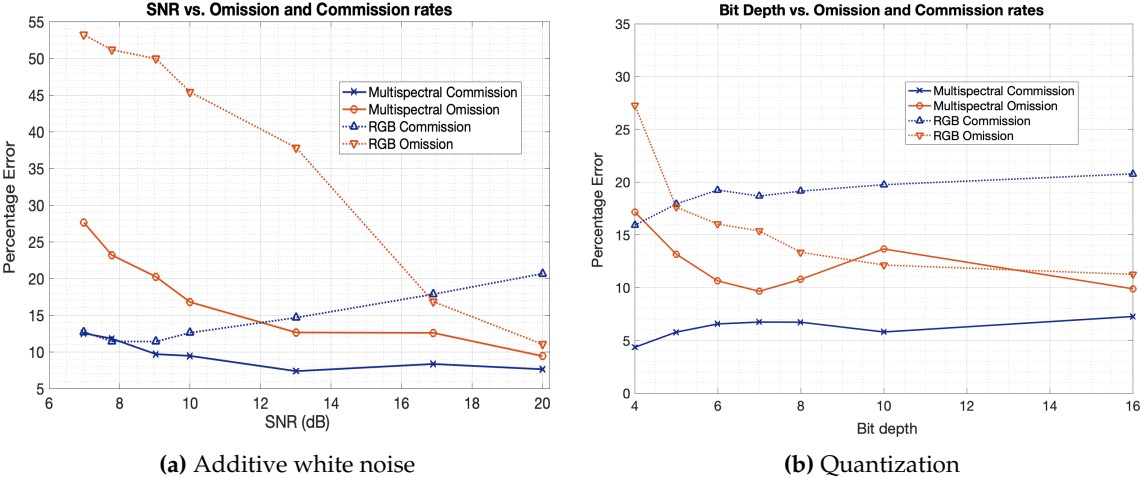

**(a)** Additive white noise      **(b)** Quantization

**Figure 5.** Omission and Commission rates of both RGB and multispectral models against the SNR (**a**) and the bit depth of the quantization (**b**). For both white noise and quantization errors of omission increase with the level of noise. Meanwhile, the effect on commission is less strong, suggesting the added noise has more of an impact on cloudy pixels than non-cloudy ones.

5.5.2. Quantization

Quantization—the reduction in bit depth—of an image, simulates a common compression technique in image processing, or a limitation in the sensor's precision. Landsat 8 imagery is delivered in 16-bit precision (although it is derived from 12-bit raw data). In our experiment, we re-quantize the Landsat 8 imagery between 16- and 4-bit precision, and track Omission and Commission rates over the Biome dataset for RGB and multispectral models. Examples of images at different bit depths show a loss of local texture information at lower precision (Figure 4b).

The effect of bit depth on error rates can be seen in Figure 5b, and exhibit similar trends for both RGB and multispectral models. In both cases, Omission rates increase at low bit rates, whilst Commission trends downwards very slightly. The multispectral results suggest a higher tolerance for quantization, with no noticeable increase in Omission until the bit depth goes below around 6-bits, whereas the RGB model begins to suffer at around 8-bits. RGB Omission increased 16.2% from 11.0% to 27.2%, significantly more than multispectral which increased 7.4% from 9.8% to 17.2%. These results suggest that performance of our cloud masking algorithm is not dependent on high precision data, requiring only around 8-bits or more to perform optimally in RGB, or less in multispectral. Additionally, this experiment supports our claim that CloudFCN learns to place high importance on textural and spatial information, because as the texture of clouds is removed and they become smoother whilst leaving pixel intensities roughly the same, the model begins misclassifying them.

## 6. Discussion

Our experiments have shown that our Deep Learning-based model performs well in a range of settings, and is immediately applicable to RGB and multispectral data as a state-of-the-art solution to the problem of cloud masking. In addition, it can be reliably used for scene-wide cloud coverage assessment, showing the potential of this algorithm for on-board use in future missions. On-board applications for convolutional algorithms are now realistic, thanks to the proliferation of several Deep Learning-specialised chips with low power requirements and small form factors. The RGB mode was used because it provided evidence for performance on the large number of RGB satellites currently being flown. Meanwhile, the multispectral mode makes our results directly comparable in a fair way with other methods that have access to these Landsat 8 bands. Of course, many satellites have different spectral band combinations, which should be the subject of further study.

In our studies multispectral bands—when available—led to better overall results. This is to be expected, given that many surface features (e.g., urban environments, exposed rock and snow) exhibit high albedo in visible bands, similar to cloud. However, the inclusion of infrared bands gives the model a larger parameter space in which to separate cloud from surface. If some of these bands are unhelpful (in the sense that they provide no correlative power for prediction of output class), then the model's weights will organise in a way such that the unhelpful bands have negligible impact on the output class.

Performance over snow and ice was reasonable in RGB Carbonite-2 data, with only 6.45% of clear images over snow or ice being marked as >50% cloud. However, on the Landsat 8 dataset performance in RGB was low over this terrain. This suggests that textural information that is useful for discrimination of cloud against snow or ice is exclusively available at higher resolutions (1 m) and is not as present at 30 m, as in Landsat 8 imagery.

The application of cloud masking over snow therefore requires further research and refinement, for which several possible directions exist. For example, simply producing more labelled cloud masks over snow may help future models perform better. Or, a preprocessing step could be used over snowy regions, either with a Deep Learning approach, or by geographical correlation with a snow-cover map, which would then be used as an extra input to the model. However, this increases the algorithm's complexity and adds dependencies for geographical information.

By applying noise to our data, we show that the algorithm can still perform reasonably well with low SNR sensors or with low bit depth data (especially for multispectral images). Although these

simulated sources of noise do not allow us to concretely extrapolate the algorithm's performance to other instruments, we believe it lends strong evidence to support the claim that this architecture will produce a robust solution for a wide range of EO cameras, assuming a relevant dataset can be sourced.

Ultimately, for Deep Learning to succeed in providing high-performance and practical tools to the EO community, greater efforts must be made by the community in creating, validating and distributing high-quality, high-volume labelled datasets. Landsat 8 has the benefit of a large, openly available cloud masking dataset which made this study (and many others) possible. However, for many satellite platforms this approach is simply not feasible, given the lack of data. In this regard, traditional, physically-informed thresholding models continue to hold a significant advantage over Deep Learning, in that they exhibit interoperability on many satellites. Future model development work should therefore focus on creating Deep Learning models that do have some form of physical information handling capabilities, in order to create interoperable models that take advantage of the high performance of Deep Learning alongside the flexibility of traditional approaches.

## 7. Conclusions

In this study, we have detailed the development of a state-of-the-art cloud masking algorithm, CloudFCN. This model is a U-Net architecture whose design includes features such as Inception modules, batch normalisation layers and Leaky ReLU layers. These are selected based on evidence of their general success in Computer Vision tasks, and our reasoning about the specific challenges of cloud masking in Remote Sensing data. Further, we provide detailed and varied performance testing, which confirms that Fully Convolutional Network architectures are indeed a powerful tool in cloud masking in a range of settings, and can perform better than previous techniques, given high-quality labelled datasets. Given the success of CloudFCN in cloud masking, we believe that the architecture is generally applicable to all segmentation tasks in Remote Sensing. Re-purposing the algorithm for new use cases is straightforward, assuming the existence of sufficient labelled data. A direct extension to this work is the masking of cloud shadows, but more tangential possibilities also exist. For example—land-use segmentation, sea ice monitoring and vegetation indexing.

The experimental results of this study underline the high accuracy of our method, but also demonstrate its robustness to noisy data and a wide range of terrains in Sections 5.4 and 5.5. This shows the algorithm's potential as a practical tool for researchers currently using Landsat 8 data, and can inform practitioners when it is most suitable to use our model and when its results should be treated with suspicion. We believe that this study not only establishes Deep Learning models as the best-performing method for cloud masking, but that it provides incentive for those working with other satellites to create large, hand-labelled datasets for segmentation problems (if they do not yet exist). This will allow Deep Learning techniques to be better exploited on a wider range of sensors.

Future work should extend this methodology to cloud shadow detection, as it is a related problem both in execution and in motivation. Similarly, our method could be extended to further distinguish between meteorological features, whether by cloud-type (e.g., cumulus or cirrus) or other features such as fogs and haze, if the labelled data for this task is created. Further research is needed to better cope with snow or icy terrain, which continues to prove the most challenging biome for our method along with others'. To advance these efforts, we have made the code used in our experiments freely available, a link to which can be found in the Supplementary Materials.

**Supplementary Materials:** The code used for the experiments conducted in this paper are available online at: https://github.com/aliFrancis/cloudFCN.

**Author Contributions:** Conceptualization, A.F., J.-P.M. and P.S.; methodology, A.F., P.S.; software, A.F.; validation, A.F. and P.S.; formal analysis, A.F.; investigation, A.F., P.S. and J.-P.M.; resources, P.S. and J.-P.M.; data curation, A.F. and P.S.; writing—original draft preparation, A.F.; writing—review and editing, J.-P.M., P.S. and A.F.; visualization, A.F.; supervision, P.S. and J.-P.M.; project administration, J.-P.M.; funding acquisition, P.S. and J.-P.M.

**Funding:** This research was funded by the UK Space Agency Centre for Earth Observation Instrumentation under OverPASS project (UKSA-CEOI-11 2018-2019) grant agreement number RP10G0435C206.

**Acknowledgments:** We thank Earth-i and SSTL for the Carbonite-2 images used in this work; We also thank USGS for generating the huge dataset of cloud masks used with Landsat-8, Cortexica for their contributions to this work, including Carbonite-2 annotations by Agneta Rusy and Antoine Broyelle for his advice; Jacqueline Campbell for her input.

**Conflicts of Interest:** The authors declare no conflict of interest.

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
