# Peer review of "CloudFCN: Accurate and Robust Cloud Detection for Satellite Imagery with Deep Learning"

_remotesensing, doi:10.3390/rs11192312_

Round 1

Reviewer 1 Report

This paper introduces a deep learning solution for cloud detection. The method, called,

CloudFCN,  is based on the  U-net architecture and fuses the shallowest and deepest layers of the network, thus routing low-level 0 visible content to its deepest layers.

The paper is well organized and written. The validation is convincing.

However, I believe that for computer vision/image processing community it would be very helpful to discuss the difference between haze/fog and cloud.  In this direction would be very interesting to discuss briefly some haze detection techniques (e.g.          Ancuti et al. A Fast Semi-inverse Approach to Detect and Remove the Haze from a Single Image. ACCV , 2010 ).

Moreover would be very helpful for the research community  that the camera ready paper to include a link to the tested images (Carbonite-2 and Landsat 8)

Author Response

This paper introduces a deep learning solution for cloud detection. The method, called, CloudFCN,  is based on the  U-net architecture and fuses the shallowest and deepest layers of the network, thus routing low-level 0 visible content to its deepest layers.

The paper is well organized and written. The validation is convincing.

We would like to thank the reviewer for their time and their comments.

However, I believe that for computer vision/image processing community it would be very helpful to discuss the difference between haze/fog and cloud. In this direction would be very interesting to discuss briefly some haze detection techniques (e.g.      Ancuti et al. A Fast Semi-inverse Approach to Detect and Remove the Haze from a Single Image. ACCV , 2010 ).

Based on the reviewer comments, we have added a paragraph related to fog/haze and the difference with clouds in the Related Work Section (Lines 87-91), containing the suggested citation, and a section of the conclusion (Lines 620-623). Unfortunately we are limited in our work by the lack of large datasets with fog/haze annotations as distinct from clouds.

Moreover would be very helpful for the research community that the camera ready paper to include a link to the tested images (Carbonite-2 and Landsat 8).

In the new version of the manuscript we have added a footnote link from which Landsat 8 can be downloaded (Line 422). On the other hand, the Carbonite-2 dataset is proprietary and unfortunately cannot be publicly released.

Many thanks.

Reviewer 2 Report

In this paper, a new deep learning-based method for cloud detection is proposed in an FCN framework. The experimental results show the proposed method achieves better performance than the competing approaches. Overall, the paper is well presented and organized, and provides a comprehensive experimental evaluation. Some concerns need to be addressed as below.

Related work section: There is a scope for improvement in this section. Listed machine learning-based methods especially existing deep learning-based methods for cloud detection concluded with less criticism/benefits/limitations etc.

Compared with other FCN-Based methods for cloud detection, what’s the significant advantages of the proposed CloudFCN model?

Cloud shadow seems to be ignored in this work. In fact, cloud and cloud shadow are inseparable for optical imagery.

The loss function for the proposed method should be specifically described.

Author Response

In this paper, a new deep learning-based method for cloud detection is proposed in an FCN framework. The experimental results show the proposed method achieves better performance than the competing approaches. Overall, the paper is well presented and organized, and provides a comprehensive experimental evaluation. Some concerns need to be addressed as below.

We would like to thank the reviewer for their care in reading our manuscript and their comments.

Related work section: There is a scope for improvement in this section. Listed machine learning-based methods especially existing deep learning-based methods for cloud detection concluded with less criticism/benefits/limitations etc.

Based on the reviewer's suggestion, we have restructured Section 2.3, adding two more recent works that employ deep learning: (Shendryk, 2019) Line 158/159, and (Li, 2018) Line 178, and rephrasing some of our comments about other works (Lines 170-185).

Compared with other FCN-Based methods for cloud detection, what’s the significant advantages of the proposed CloudFCN model?

As mentioned individually at different points of the manuscript, the main contribution of our approach in relation to other fully convolutional methods are (1) the use of a ’receptive field’ as a determining factor of the network size (2) the use of InceptionNet-style modules within the network and (3) the (tailored to imbalanced classes) class-weighting scheme. In addition to these design choices, the experimental validation is unique for Deep Learning approaches, as it combines multiple performance metrics, and tests across many variables (e.g. terrain, noise).

In order to make these contributions more clear to the reader, we have added a clarifying paragraph (Lines 61-74).

Cloud shadow seems to be ignored in this work. In fact, cloud and cloud shadow are inseparable for optical imagery.

We totally agree with the reviewer about the connection between clouds and cloud shadows. However, the datasets used in this work either do not have cloud shadow annotations at all (Carbonite-2) or have sparse shadow annotations, omitting many of the cloud shadows in the dataset (Landsat 8). This sparsity of annotation is not an issue for physical thresholding methods, however training of a Deep Learning algorithm is difficult with inconsistent annotations.

We have added an explanation of this in the Results section (Lines 398-400).

The loss function for the proposed method should be specifically described.

Interestingly, we didn’t find significant performance divergence among several commonly used loss functions. However, based on the reviewer comments we have stated the loss functions used (Lines 315-318).

Many thanks.

Reviewer 3 Report

Please see attached doc.

Author Response

Dear reviewer,

Thanks for the very quick and helpful review!

We think you raise some very useful points, but would like to seek clarification/revision on a couple of them, and suggest some possibilities for addressing multiple points at once given that they are inter-related. Please see the attachment for our thoughts. We will proceed to make the changes once we have agreed with you a productive method for addressing all your issues.

Many thanks.

Round 2

Reviewer 2 Report

The authors have made a good revision, and I suggest publishing it as the present form.

Reviewer 3 Report

Manuscript is improved significantly and i have no more suggestions.